# End-of-life experiences in individuals with dementia with Lewy bodies and their caregivers: A mixed-methods analysis

Easton Wollney[1], Kaitlin Sovich[2,3], Brian LaBarre[4], Susan M. Maixner[5], Henry L. Paulson[6], Carol Manning[7], Julie A. Fields[8], Angela Lunde[9], Bradley F. Boeve[9], James E. Galvin[10], Angela S. Taylor[11], Zhigang Li[4], Hannah J. Fechtel[2,3], Melissa J. Armstrong[2,3]*

1 Department of Health Outcomes and Biomedical Informatics, University of Florida College of Medicine, Gainesville, Florida, United States of America, 2 Department of Neurology, University of Florida College of Medicine, Gainesville, Florida, United States of America, 3 Norman Fixel Institute for Neurological Diseases, Gainesville, Florida, United States of America, 4 Department of Biostatistics, University of Florida College of Medicine, Gainesville, Florida, United States of America, 5 Department of Psychiatry, University of Michigan, Ann Arbor, Michigan, United States of America, 6 Department of Neurology, University of Michigan, Ann Arbor, Michigan, United States of America, 7 Department of Neurology, University of Virginia, Charlottesville, Virginia, United States of America, 8 Department of Psychiatry and Psychology, Mayo Clinic, Rochester, Minnesota, United States of America, 9 Department of Neurology, Mayo Clinic, Rochester, Minnesota, United States of America, 10 Department of Neurology, Comprehensive Center for Brain Health, University of Miami Miller School of Medicine, Miami, Florida, United States of America, 11 Lewy Body Dementia Association, Lilburn, Georgia, United States of America

* Melissa.Armstrong@neurology.ufl.edu

## Abstract

### Background

Dementia with Lewy bodies (DLB) is one of the most common degenerative dementias, but research on end-of-life experiences for people with DLB and their caregivers is limited.

### Method

Dyads of individuals with moderate-advanced DLB and their primary informal caregivers were recruited from specialty clinics, advocacy organizations, and research registries and followed prospectively every 6 months. The current study examines results of caregiver study visits 3 months after the death of the person with DLB. These visits included the Last Month of Life survey, study-specific questions, and a semi-structured interview querying end-of-life experiences.

### Results

Individuals with DLB (n = 50) died 3.24 ± 1.81 years after diagnosis, typically of disease-related complications. Only 44% of caregivers reported a helpful conversation with clinicians regarding what to expect at the end of life in DLB. Symptoms commonly worsening prior to death included: cognition and motor function, ADL dependence, behavioral features, day-time sleepiness, communication, appetite, and weight loss. Almost 90% of participants received hospice care, but 20% used hospice for <1 week. Most caregivers reported overall

**Data Availability Statement:** All relevant data are within the paper and its Supporting Information files. Individual participant demographic data and descriptive quantitative data presented in this paper

are available in the S1 Table. The codebook with relevant excerpts is available in S3 File to supplement the quotes provided in the paper.

**Funding:** This work was supported by the National Institutes on Aging grant number R01AG068128. Research reported in this publication was supported by the University of Florida Clinical and Translational Science Institute (hosting REDCap), which is supported in part by the NIH National Center for Advancing Translational Sciences under award number UL1TR001427. The content is solely the responsibility of the authors and does not necessarily represent the official views of the National Institutes of Health/National Institutes on Aging. The funders had no role in study design, data collection and analysis, decision to publish, or preparation of the manuscript.

**Competing interests:** I have read the journal's policy and the authors of this manuscript have the following competing interests: EW, BL, SMM, ZL, and HJF report no competing interests. HLP: HLP receives funding from the NIA (1P30AG053760) and is the local PI of a Lewy Body Dementia Association Research Center of Excellence. CM: CM receives research support from ACL/DHHS (90ALGG0014-01-00), NIA/NIH (2SB1AG037357-04A1, R01-AG-054435), HRSA (U1QHP287440400) and DoD (AZ190036). She is the local PI of a Lewy Body Dementia Association Research Center of Excellence. JAF: JAF receives research support from the NIA (U01NS100620, R01AG068128, R43AG65088). AL: AL receives research support from the NIA (P30AG62677, R43AG65088). She is a Program Coordinator for the local a Lewy Body Dementia Association Coordinating Center Research Center of Excellence. BFB: BFB has served as an investigator for clinical trials sponsored by Biogen, Alector, and EIP Pharma. He serves on the Scientific Advisory Board of the Lewy Body Dementia Association, Association for Frontotemporal Degeneration, and Tau Consortium. He is the site PI of a Lewy Body Dementia Association Research Center of Excellence program, as well as coordinating center PI of the program. He receives research support from the NIH, the Mayo Clinic Dorothy and Harry T. Mangurian Jr. Lewy Body Dementia Program, and the Little Family Foundation. JEG: JEG is the creator of the QDRS and the LBCRS. He is supported by grants from the National Institutes of Health (R01 AG069765, R01 AG057681, R01 NS101483, P30 AG059295, U54 AG06354, R01 AG056531, U01 NS100610, R01 AG056610, R01 AG054425, R01 AG068128) and the Leo and Anne Albert Charitable Trust. He is the local PI of the Lewy Body Dementia Association Research Center

positive experiences in the last month of life, but this was not universal. Having information about DLB and what to expect, access to support, and hospice care were healthcare factors associated with positive and negative end of life experiences. Hospice experiences were driven by communication, care coordination, quality care, and caregiver education.

## Conclusion

Most caregivers of individuals who died with DLB reported positive end-of-life experiences. However, the study identified multiple opportunities for improvement relating to clinician counseling of patients/families, support/hospice referrals, and monitoring individuals with DLB to identify approaching end of life. Future research should quantitatively identify changes that herald end of life in DLB and develop tools that can assist clinicians in evaluating disease stage to better inform counseling and timely hospice referrals.

## Trial registration

Trial registration information: NCT04829656.

## Introduction

Dementia with Lewy bodies (DLB) is one of the most common degenerative dementias after Alzheimer disease (AD) dementia. Individuals with DLB have dementia accompanied by features including fluctuating alertness, visual and non-visual hallucinations, REM sleep behavior disorder, parkinsonism, excessive daytime sleepiness, autonomic dysfunction, postural instability, falls, delusions, anxiety, depression, and apathy [1]. The average survival of individuals with clinically diagnosed DLB (from diagnosis to death) is only 4.11 years (SD ± 4.10), shorter than survival in individuals with AD dementia (5.66 ± 5.32 years, p<0.01) [2]. Reported survival is longer in pathologic cohorts (i.e., with post-mortem examination) [3] or when calculating time from symptom onset [4]. Based on a prior survey (n = 658), the most common cause of death in DLB is failure to thrive (65%), followed by pneumonia/swallowing difficulties (23%), medical conditions other than pneumonia (19%), and falls or complications from a fall (10%) (multiple categories allowed) [4].

Despite the facts that DLB is common and most individuals with DLB die of disease-related complications, relatively little is known regarding the end-of-life experience in DLB. In a 2017 survey conducted through the Lewy Body Dementia Association (n = 658), only 40% of family members of individuals with DLB said their physician discussed what to expect at the end of life in DLB. Only 22% reported this was discussed to a helpful degree [4]. While almost 80% of respondents reported that the person with DLB received hospice, a quarter of respondents felt that hospice was involved too late. Fewer than half of respondents felt prepared for what to expect [4]. These findings were echoed in subsequent interviews with caregivers of individuals who died with DLB. Multiple caregivers reported that physicians never discussed that DLB can be terminal or what to expect [5]. Caregivers also described not understanding hospice or how it could help and said that their physicians were unaware of hospice as an option or how to arrange it [5].

The ongoing PACE-DLB (Predicting ACcurately End-of-Life in Dementia with Lewy Bodies and Promoting Quality End-of-Life Experiences) study aims to investigate predictors of end-of-life in DLB to better inform counseling of patients and families [6]. The current mixed-

of Excellence at the University of Miami and serves on the Scientific Advisory Board of the Lewy Body Dementia Association. AST: AST is an employee of the Lewy Body Dementia Association. MJA: MJA receives research support from the NIH (R01AG068128, P30AG066506, R01NS121099, R44AG062072), the Florida Department of Health (grants 20A08, 24A14, 24A15), and as the local PI of a Lewy Body Dementia Association Research Center of Excellence. She serves on the DSMBs for the Alzheimer's Therapeutic Research Institute/ Alzheimer's Clinical Trial Consortium and the Alzheimer's Disease Cooperative Study. She has provided educational content for Medscape, Vindico CME, and Prime Inc. This does not alter our adherence to PLOS ONE policies on sharing data and materials.

methods analysis investigates end-of-life experiences for the first 50 study participants who died during the study as reported by caregivers at the 3-month post-death visit. This study repeats some of the 2017 LBDA survey and interview questions but also prospectively collects data regarding the participants with DLB and their caregivers and performs additional questionnaires approximately 3 months after the death of the person with DLB.

## Methods

### Study design and recruitment

PACE-DLB is an observational longitudinal cohort study involving five Lewy Body Dementia Association Research (LBDA) Research Centers of Excellence in the United States (University of Florida, University of Michigan, Mayo Clinic Rochester, University of Virginia, University of Miami) and recruitment of a virtual cohort (targeting individuals not receiving subspecialty care). Recruitment for the virtual cohort occurred through the LBDA, Lewy Body Dementia Research Center, clinicaltrials.gov, Alzheimer's Association Trial Match, the Fox Trial Finder, advertising in caregiver newsletters, support group presentations, and unaffiliated neurologists who learned about the study and offered to share recruitment materials.

### Study participants

Inclusion criteria were: 1) person with a clinical diagnosis of DLB (reported by person with DLB, caregiver, or clinician and confirmed with a Lewy Body Composite Risk Score [LBCRS] [7] score $\geq$3), 2) person with DLB and primary informal caregiver willing to participate as a dyad, 3) U.S. residents, 4) at least moderate severity dementia as assessed by the Quick Dementia Rating System (QDRS, score of >12 suggestive of at least moderate dementia) [8], 5) person with DLB expected to live at least 6 months (by clinician or participant estimation), and 6) caregiver Telephone Interview for Cognitive Status score of >31 [9] to ensure that the caregiver was able to reliably complete study visits. The participating caregiver was the person providing the majority of the informal care for the person with DLB and attending the majority of their clinical visits. The current analysis includes the first 50 dyads with a post-death study visit. Because saturation of themes was reached with this cohort, the interview questions subsequently changed. Thus, the current analysis includes visits relating to the original semi-structured interview. Included post-death visits reflect participants who died after the baseline visit and before the 6-month follow up and participants who had multiple study visits prior to death.

### Standard protocol approvals, registrations, and patient consents

This study was first approved by the University of Florida institutional review board (IRB202001438) on 6/23/2020. It was registered on clinicaltrials.gov (NCT04829656). Caregivers provided written informed consent for their own participation and individuals with DLB provided consent or assent with additional consent from a legally authorized representative. The first participant was enrolled on 2/25/2021. The last visit included in the current analysis of the first 50 post-death visits occurred on 7/20/2023. The semi-structured interview changed after this time due to saturation of themes for all interview questions. Study enrollment closed 12/22/2023; follow up visits are ongoing.

### Study visits and measures

Study visits were designed for virtual and in-person completion given a priori plans for inclusion of a virtual cohort. Because funding occurred during the COVID pandemic, dyads

enrolled at LBDA Research Centers of Excellence were allowed to choose in-person or virtual study completion. Virtual study visits were completed by phone or HIPAA-compliant video-conferencing. Measures were administered verbally by study coordinators to minimize differences in conduct between virtual and in-person visits and because some older caregivers could be uncomfortable with electronic form completion. Coordinators entered data directly into electronic case report forms using REDCap (Research Electronic Data Capture) tools hosted at the University of Florida [10,11]. Visits occurred every 6 months until the death of the person with DLB. A post-death visit was completed with the caregiver approximately 3 months after the death of the person with DLB and this visit formed the basis for the current analysis. This visit included questions about the end of life derived from the prior DLB survey [4], the Last Month of Life Survey from the National Health and Aging Trends Study [12], and a semi-structured interview (S1 File) that was based on the prior interview study [5] but revised for PACE-DLB with input from the study team, including a former caregiver. Interviews were scheduled to last up to 30 minutes. The current analysis focuses on the first research question (end-of-life experiences); interviews for the second research question about post-death experiences are ongoing. Interviews were performed by the research coordinator who had conducted prior study visits to build on established relationships. Coordinators received training on conducting interviews as part of site orientation. The part of the study visit including the Last Month of Life Survey and semi-structured interview was audio recorded with participant consent.

## Analysis

Descriptive statistical measures included means, standard deviations, minimum and maximum values of continuous variables (range) and frequency of categorical variables. The current analysis focused on qualitative experiences and no quantitative analyses were performed with the descriptive and qualitative results. For this qualitative analysis, there were two research questions: 1) what were the experiences of individuals with DLB at the end of life (as reported by the caregiver) and 2) what healthcare factors were associated with positive or negative experiences toward the end of life, particularly for caregivers? Thematic analysis was used to analyze and present interview data through an inductive process in which themes emerged from the data, with no theoretical constructs or existing framework used to classify data [13,14]. Braun & Clarkes' criteria for thematic analysis [13] involves an iterative process beginning with broad codes, then collapsing codes, and continuously refining codes until themes are finalized and contain rich description [15]. This process overlaps with the constant comparative method commonly used in grounded theory. However, thematic analysis does not require theory development [13]. An iterative process ensured rigor [16]. Themes were considered saturated when no new codes emerged, utilizing an inductive thematic approach [17].

Prior qualitative analyses relating to end of life in DLB were reviewed to inform coding [5,18,19] but a new codebook was developed for the current analysis. One coder (ENW, a post-doctoral researcher with experience in qualitative methods) open-coded three transcripts to develop an initial coding scheme. The codebook was then created by two researchers (ENW, MJA) and subsequently refined throughout the process. Codes were collapsed by the primary analysis team (ENW, MJA) into broader categories representing related concepts and overarching themes [13]. After overarching themes were developed, the primary coder (ENW) closed-coded remaining transcripts to further establish thematic descriptions and exemplars. The codebook was then shared with co-authors for further verification. Participant checking was not performed. Qualitative data were managed using ATLAS.ti software. Consolidated criteria for reporting qualitative research guided the reporting of study findings (S2 File) [20].

## Results

Individuals with DLB (n = 50) died a mean 3.24 years (SD 1.81, range 1–8) after diagnosis (Table 1).

Most of the caregivers were women (84%). Caregivers included 31 wives (62%), 11 daughters (22%), and 8 husbands (16%). Twenty-eight (56%) of the individuals with DLB died at home, 9 (18%) in a specialty memory care facility, 5 (10%) in a hospice facility, 3 (6%) in a

**Table 1. Baseline characteristics (at time of last pre-death visit).**

| | Individuals with DLB (n = 50) | Caregivers (n = 50) |
|---|---|---|
| Age at last pre-death visit (years; mean, SD, range) | 77.2 ± 6.92 (62–97) | 67.28 ± 9.2 (38–82) |
| Biologic Sex (% male) | 37 (74%) | 8 (16%) |
| Gender Identity (% male) | 36 (72%) | 8 (16%) |
| Race (n, %) | | |
| White | 48 (96%) | 47 (94%) |
| Black or African-American | 1 (2%) | 1 (2%) |
| Native Hawaiian or other Pacific Islander | 0 | 1 (2%) |
| Unknown or not reported | 1 (2%) | 1 (2%) |
| Ethnicity (n, %) | | |
| Non-Hispanic | 49 (98%) | 48 (96%) |
| Hispanic | 1 (2%) | 2 (4%) |
| Highest education | | |
| Doctorate or professional degree | 7 (14%) | 5 (10%) |
| Master's degree | 6 (12%) | 13 (26%) |
| Bachelor's degree | 13 (26%) | 20 (40%) |
| Associate degree | 6 (12%) | 3 (6%) |
| Technical/occupational certificate or some college | 12 (24%) | 7 (14%) |
| High school or equivalent | 5 (10%) | 2 (4%) |
| 8th grade or less | 1 (2%) | 0 |
| Time from diagnosis to death (years; mean, SD, range) | 3.24 ± 1.81 (1–8) | N/A |
| Caregiver role | N/A | |
| Spouse | | 39 (78%) |
| Child | | 11 (22%) |
| Years spent caregiving (mean ± SD, range) | N/A | 3.9 ± 2.22 (1–10) |
| Hours per day "on duty" for caregiving (mean ± SD, range) | N/A | 15.08 ± 9.45 (0–24) |
| Hours per day must be in same room as loved one (mean ± SD, range) | N/A | 8.58 ± 8.31 (0–24) |
| Live with patient (yes, %, n) | N/A | 35 (70%) |
| Anyone else in home (yes, %, n) | N/A | 9 (24.3%) |
| Location where person with DLB lived (pre-death visit) | | N/A |
| Home | 37 (74%) | |
| Long term care facility/nursing home | 9 (18%) | |
| Assisted living | 4 (8%) | |
| Residential location | | N/A |
| Suburban | 29 (58%) | |
| Rural | 13 (26%) | |
| Urban | 8 (16%) | |
| Type of medical providers seen (could select >1) | | N/A |
| Primary care provider or geriatrician | 41 (82%) | |
| General neurologist | 16 (32%) | |
| Specialist (dementia, movement disorders, or DLB) | 26 (52%) | |
| Cohort | | N/A |
| Center of Excellence cohort | 17 (34%) | |
| Virtual recruitment | 33 (66%) | |

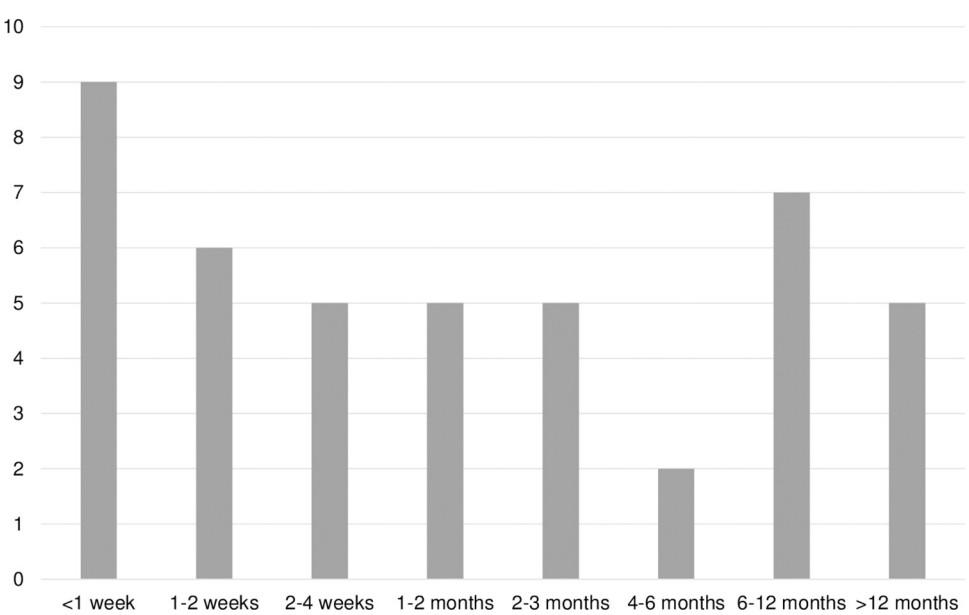

**Fig 1. Duration of hospice care prior to death (n = 46).**

long term care facility, 3 (6%) in the hospital, and 2 (4%) in assisted living. Most participants (94%) had advance directives in place and 44 (88%) were enrolled in hospice prior to death. For individuals using hospice resources (n = 46), the median duration was 1–2 months, but 20% had hospice resources for <1 week and 43% had hospice for <1 month (Fig 1). While most caregivers felt that hospice was initiated at the right time (31/46, 70.5%), nine (20%) felt hospice services were started too late (2: too early, 2: unsure).

Only 27 caregivers (54%) reported that health care professionals discussed what to expect at the end of life in DLB and only 22 (44%) said that they felt that this discussion was helpful. Death was reported to be expected in 31 cases (62%). Caregivers most commonly reported that individuals with DLB died from the DLB itself (21, 42%; exact details unknown) or failure to thrive (9, 18%). Thirteen caregivers reported that the person with DLB died of medical causes (e.g. stroke, coronary artery disease, congestive heart failure, kidney failure, COVID), with 3 of these also identifying DLB as a cause of death. Some caregivers reported that they were told the person with DLB died of "natural causes" or simply "stopped breathing;" four did not specify the cause of death. One caregiver reported the person with DLB died from trouble swallowing and another reported death relating to complications from a fall.

### Experiences in the last month of life

Thirty-nine (78%) caregivers reported that the person with DLB received either very good or excellent care in the last month of life (Table 2). Some caregivers, however, reported fair or poor experiences. When pain, trouble breathing, or anxiety/sadness were present at the end of life, most caregivers reported that these symptoms were adequately treated. Decisions were usually made with input from the person with DLB and their family and were congruent with what the person with DLB would have wanted, though this was not universal (Table 2). People with DLB were typically treated with respect based on caregiver observations. Over 90% of caregivers reported that they were usually or always kept informed about what was happening during the last month of life, though a couple caregivers reported they were not kept informed at all. Health professionals spoke to the person with DLB about his or her religious beliefs less

**Table 2. Responses to the last month of life survey.**

| Section | Question | Response (total n = 50) |
|---|---|---|
| **Rate care** | Overall, how would rate [SP]'s care in the last month of life? Would you say it was, excellent, very good, good, fair, or poor? | Excellent: 22 (44.0%)<br>Very good: 17 (34.0%)<br>Good: 2 (4.0%)<br>Fair: 6 (12.0%)<br>Poor: 3 (6.0%) |
| **Pain** | During the last month of [SP]'s life, were there times when [he/she] experienced pain? | Yes: 35 (70.0%)<br>No: 14 (28.0%)<br>Don't know: 1 (2.0%) |
| | Did [SP] get any help in dealing with [his/her] pain? | Yes: 32/35 (91.4%)<br>No: 3/25 (8.6%) |
| | How much help in dealing with [his/her] pain did [SP] receive: less than was needed, more than was needed, or about the right amount? | Less than was needed: 5/32 (15.6%)<br>More than was needed: 1/32 (3.1%)<br>About the right amount: 26/32 (81.3%) |
| **Breathing trouble** | During the last month of [SP]'s life, were there times when [he/she] had trouble breathing? | Yes: 26 (52.0%)<br>No: 24 (48.0%) |
| | Did [SP] get any help in dealing with [his/her] trouble breathing? | Yes: 20/26 (76.9%)<br>No: 6/26 (23.1%) |
| | How much help in dealing with [his/her] breathing did [SP] receive: less than was needed, more than was needed, or about the right amount? | Less than was needed: 1/20 (5.0%)<br>About the right amount: 17/20 (85.0%)<br>Don't know: 2/20 (10.0%) |
| **Anxious or sad** | During the last month of [SP]'s life, did [he/she] have any feelings of anxiety or sadness? | Yes: 33 (66.0%)<br>No: 10 (20.0%)<br>Don't know: 6 (12.0%)<br>Refused: 1 (2.0%) |
| | Did [SP] get any help in dealing with [his/her] feelings of anxiety or sadness? | Yes: 25/33 (75.8%)<br>No: 8/33 (24.2%) |
| | How much help in dealing with these feelings did [SP] receive: less than was needed, more than was needed, or about the right amount? | Less than was needed: 5/25 (20.0%)<br>About the right amount: 20/25 (80.0%) |
| **Decisions about care without input** | During the last month of [SP]'s life, was there ever a decision made about [his/her] care or treatment without enough input from [him/her] or [his/her] family? | Yes: 4 (8.0%)<br>No: 45 (90.0%)<br>Does not apply/no care in last month of life: 1 (2.0%) |
| **Decisions about care not wanted** | During the last month of [SP]'s life, was there any decision made about care or treatment that [he/she] would not have wanted? | Yes: 6 (12.0%)<br>No: 44 (88.0%) |
| **Personal care needs met** | During the last month of [SP]'s life, how often were [his/her] personal care needs, such as bathing, dressing, and changing bedding, taken care of as well as they should have been: always, usually, sometimes, or never? | Always: 33 (66.0%)<br>Usually: 9 (18.0%)<br>Sometimes: 8 (16.0%)<br>Never: 0 |
| **Treated with respect** | During the last month of [SP]'s life, how often were [he/she] treated with respect by those who were taking care of [him/her]: always, usually, sometimes, or never? | Always: 39 (78.0%)<br>Usually: 8 (16.0%)<br>Sometimes: 2 (4.0%)<br>Never: 0<br>Don't know: 1 (2.0%) |
| **Informed about condition** | During the last month of [SP]'s life, how often were you or other family members kept informed about [him/her] condition: always, usually, sometimes, or never? | Always: 40 (80.0%)<br>Usually: 7 (14.0%)<br>Sometimes: 1 (2.0%)<br>Never: 2 (4.0%) |

(*Continued*)

**Table 2.** (Continued)

| Section | Question | Response (total n = 50) |
|---|---|---|
| More than one doctor | During the last month of [SP]'s life, was there more than one doctor involved in [his/her] care? | Yes: 28 (56.0%)<br>No: 21 (42.0%)<br>Does not apply/no care in last month of life: 1 (2.0%) |
| | During the last month of [SP]'s life, was it always clear to you which doctor was in charge of [his/her] care? | Yes: 20/28 (71.4%)<br>No: 7/28 (25.0%)<br>Don't know: 1/28 (3.6%) |
| Talk about religious beliefs | During the last month of life, did any doctors, nurses, other health professional talk with [SP] about [his/her] religious beliefs? | Yes: 24 (48.0%)<br>No: 21 (42.0%)<br>Don't know: 3 (6.0%)<br>Does not apply/no care in last month of life: 2 (4.0%) |
| | During the last month of [SP]'s life, do you think [he/she] had as much contact of this kind as [he/she] wanted? | Yes: 19/24 (79.2%)<br>No: 3/24 (12.5%)<br>Don't know: 2/24 (8.3%) |

SP: Study participant (name inserted during interview). This survey used branching logic so the total "n" for follow up questions is based on the responses to prior questions. The relevant "n" for each response is noted in the response column.

than half the time, but when this occurred, caregivers felt the degree of contact was usually in line with what the person wanted.

## Interview themes for end-of-life experiences

**Worsening symptoms at the end of life.** When relating experiences of the person with DLB towards the end of life, caregivers described both physical and cognitive declines:

"I felt like. . .his mind was gone several months before his physical self. So, when he really started going downhill physically, not being able to bear his weight, being incontinent, needing help holding a utensil—when those became apparent and got worse, then I felt like we were approaching kind of that last road." (1927–31, wife)

Caregivers described that when approaching the end of life, the person with DLB was increasingly unable to complete ADLs such as eating, toileting, bathing, and mobility, and also had increased trouble communicating (Table 3). Cognitive, behavioral, and psychiatric symptoms commonly worsened preceding death. Other symptoms included increased sleepiness, loss of appetite, weight loss, and breathing changes (Table 3).

## End-of-life trajectories

Multiple caregivers described being surprised by a sudden and rapid worsening leading to the death of the person with DLB:

"I just didn't expect it had to happen so quickly because he was walking and talking, then the next week he wasn't able to really do anything. . ." (1923–01, wife)

"I found it shocking and painful that two weeks before his death, he was walking around, and we were doing things together and I would—I was feeding him. . .and we would sit and watch TV together and things like. That seemed kind of like a normal time. And then two weeks later he was done. It still shocks me that [he] went so quickly." (1925–06, wife)

**Table 3. Experiences of individuals with dementia with lewy bodies prior to death: Exemplar quotes.**

| Loss of motor skills/functions | |
|---|---|
| **Eating and feeding** | "I think in the last three months, definitely he was in a steady decline... It was harder for him to eat on his own. I had to start feeding him, more and more consistently as well. He might have a day where he could actually pick up the utensils and feed himself. And then some days he just wouldn't." (1927–100, daughter) |
| **Toileting and bathing** | "The incontinence was a lot to deal with, but the dementia wasn't so bad that he couldn't keep his Depends on. . .he knew the number one rule in this house was to wear Depends." (1927–44, wife)<br>"I had to wash his hair and dry it, and I would dry him off, help him get dressed." (1926–01, wife) |
| **Mobility** | "Not being able to walk was a big deal. And then he started having balance problems. Even sitting in a chair, he would tip to one side or the other, and of course the lack of his fine motor movement, that was a problem too. Those symptoms made it very difficult because he couldn't do all the things that he always did." (1927–10, wife)<br>"She was getting to the point where she was going to need a two person assist. . . one of the last times I saw her, she needed two people to get her from the wheelchair that she was sitting in to a chair, an easy chair, which was like a foot away, not even." (1927–68, daughter) |
| **Speech and communication** | "The [disease] started taking away her voice. It was hard to get her to say anything and when she said anything, it was hard to hear her because her voice had become so soft." (1927–83, husband)<br>"She wasn't very communicative. . . I can't pinpoint a time, I can't say it was [in the last] three months or that, but she did become less sort of aware. You know, wasn't talkative, and she wasn't facially communicating as much." (1927–22, husband) |
| Cognitive, behavioral, and psychiatric symptoms | |
| **Cognitive decline, fluctuations** | "I noticed she didn't understand me the way she understood me prior. She wouldn't, uh, simple [things] like, are you going to bed? She didn't understand it. The English, it's like she lost it." (1927–17, daughter)<br>"We could have a really wonderful conversation. . . and then he would stand up and I'd say, 'What's wrong?' And he'd say, 'I have to pee. Where's the bathroom?' We had just been having this amazing deep conversation and then once he stood up, everything disappeared, and he didn't know where he was." (1927–44, wife)<br>"There were a couple of incidences where he would—he would just—It was almost like he went into this zombie mode where—Well, I thought he was going to fall out of his chair but, his eyes were wide awake. . . it scared me so bad. I just kept shaking him. . . and finally [he] blinked his eyes." (1924–16, wife) |
| **Psychosis (hallucinations, delirium, paranoia)** | "She previously had pretty commonly had hallucinations of cats and dogs and sometimes babies, which she enjoyed, [She] found them cute and comforting, I think.. . .[but] about eight days before she died, she started expressing seeing a man in her room." (1927–12, daughter)<br>"That last month, there was a lot of confusion, delirium." (1927–25, wife) |
| **Anxiety** | "He was getting more difficult to deal with––his anxiety and paranoia. They were getting much worse." (1926–01, wife)<br>"His anxiety was—it was more related to if I was out of his presence. . . of course, his sense of timing was not good anymore either. So, a minute to him was a long time . . . I could be sitting across the room from him, but out of his line of sight, he would drift off to sleep, wake up, and have anxiety because he didn't know where I was." (1923–06, wife) |
| **Agitation, behavior changes** | "The last days, he was in the emergency room, and he was really just in terrible shape, angry, just totally out of control, both physically and mentally. It was just totally out of character, his violence, his yelling at people. He was punching at people, kicking, and all of that. Ultimately, I know they gave him morphine at hospice. They were trying to sedate him, calm him down. So, it was an agonizing demise." (1926–06, wife)<br>"He became violent, destructive. And so it was like he died before he died and that is excruciating. . . if the person is violent and he has not been violent before, it just crushes you. So, it crushes your spirit to see that because that's not the person that you knew before." (1927–40, wife) |

*(Continued)*

**Table 3.** (Continued)

| Social withdrawal and depression | "There was a change in the last specifically week. It seemed like overnight he became even more withdrawn, and it seemed hard to believe because he was already withdrawn, but we just knew there was something really off. . . . He just seemed like he was not participating in anything. He just wanted to lay down in bed all the time." (1924–04, wife)<br>"One of his favorite things was singing, so we would go to the entertainment every day that they had there and he would sing along. And I noticed maybe the last week or so that he wasn't singing and that was his favorite thing. . . [the nurse] was very alarmed, she thought, you know, this is not like him." (1927–07, wife) |
|---|---|
| **Other symptoms** | |
| **Excessive sleepiness** | "He's always slept a lot, but that last week just slept so [much]. . . . very difficult to get him to even take a pill and then he was gone [asleep] again." (1927–23, wife)<br>"Really the last month he was pretty much just sleeping, and he would occasionally sort of wake up, but he was pretty much out of it by then." (1927–60, wife) |
| **Loss of appetite, weight loss** | "We backed off of a lot of things. We backed off trying to feed him. He clearly had lost his appetite. . ." (1927–23, wife)<br>"He was so thin. He was eating very little at the end and we were giving him Ensure, everything to keep him. . .strong enough." (1924–04, wife)<br>"Definitely the weight loss [was a sign]. How he was eating, it was so minimal." (1927–59, wife) |
| **Breathing changes** | "Near the end, he was breathing pretty fast, according to my son, and so they would give him whatever medication they gave him and that would slow down his breathing a little bit more." (1927–07, wife)<br>"Those last couple of days were very trying, with her laboring to breathe. The last day, we got to the point where we were injecting the morphine orally in the corner of her mouth every 2–3 hours because of the labor intensity of her breathing." (1923–12, husband) |

"It was a shock to me. And of course, it was shock to the family because we didn't––my anticipation of his life was going to be that I would end up having to have in home care. . . [but] he was still eating everything I gave him. He had many falls and of course the neurologist suggested that, 'this is what we're going to be going down.' . . .none of that happened. Just that quick, he died. So, I don't know what he died of. I don't think it was the dementia that caused his death, [or] whether something else happened." (1926–09, wife)

Other caregivers described variability in the person with DLB's trajectory prior to death: "he could have a bad couple of weeks and then kind of have a good few days; so that part, I think to me was the unpredictable part" (1927–100, daughter). "And the week before he died, that's the time when he would rally back and I thought, well, maybe he's going to come off hospice. . . then he passed away" (1927–53, daughter).

## Healthcare factors associated with positive or negative experiences

Healthcare factors associated with positive or negative experiences when the person with DLB was dying related to (1) having information about DLB and what to expect, (2) access to support, and (3) utilizing hospice care.

Having information about DLB and what to expect. Caregivers wanted to receive DLB-related education, including information about the disease itself, what to expect at the end of life, and how to provide care to someone with DLB. Caregivers often described having to search for this information themselves. Regarding education about DLB, caregivers expressed

wanting more information and encouraged others to self-educate: "I say this to even people beginning the journey, especially with Lewy body, is read. . . read what you can read [from] the LBDA, ask your doctor" (1927–10, wife). However, as the same caregiver pointed out, "Some doctors need to be educated because they don't know everything, especially primary care" (1927–10, wife). Caregivers particularly desired information about what to expect at the end of life for individuals with DLB, especially since hospice was sometimes unable to provide this information. "I wish that I had knew more about kind of what to expect. . .I don't know if all the patients go into delirium like that. I just wished that I had been better prepared to have recognized what was going on. Even hospice didn't know" (1927–25, wife). Other caregivers, however, felt intimidated when they read about what to expect at the end of life: "When I read about people's experiences, I was terrified and anticipating the difficulty swallowing and a long period of immobility" (1927–01, ex-wife).

Caregivers also wanted caregiving advice and tips. For example, one caregiver highlighted the helpfulness of recommendations to make dressing easier, such as using Velcro shoes or shirts: "There are little improvements that go a long way and meet him where he's at" (1927–44, wife). Others discussed information about psychosocial aspects of caregiving: "I was reading an article about the burden on caregivers for LBD. One element of that burden being the sense of being not fully competent. And I was thinking, how would it ever be possible to feel competent when you're walking into a room and this person that you love is saying, 'I can't find my fingers'" (1927–01, ex-wife). Often advice came from internet searches or support groups rather than healthcare teams. "I had a support group for people whose [loved ones] have dementia. . . They understood and also, they were an incredible resource for material. You know, I got two wheelchairs from, I got two walkers, different walkers, and just information about, 'well, what kind of diapers did you use? And how did you deal with this?'. . . just tips on how to deal with the many things" (1927–75, wife).

Access to support. Caregivers discussed the importance of both formal support (e.g. support groups) and support from the community (e.g. family, friends, neighbors, other caregivers). Caregivers recommended attending support groups as way to learn coping techniques, vent frustrations, and "hear the experiences of other people, and then you don't feel as alone because you find out that there are similarities, and when you find those similarities, it eases the burden" (1926–01, wife). Attending support groups taught caregivers about ways to cope: "People are really creative [with] the ways they find to get through things and handle things" (1923–10, wife). As noted above, support groups were also a place for caregivers to find information that they were not receiving from their medical teams: "I learned more about Lewy body from being on the Facebook support groups for Lewy body caregivers. I learned more about it there than any place else" (1927–77, wife).

Caregivers also described fostering relationships and community as a way to obtain support from family, friends, and neighbors:

"Probably the biggest advice is, you need to surround yourself with friends and family because you can't go through it alone. You have to have your support–people you can just vent to, people that care for you and what you're going through. . . those connections and relationships are so important" (1927–03, husband).

"I think it's really important to nurture the relationships you have with the people that you have, [so that] you have a support system. . . A lot of people I know don't have very many friends or aren't very involved. . . or don't get along with their neighbors. They don't go out, they're somewhat isolated, and I think you have to be as involved with as many people as you can" (1923–10, wife).

Sometimes such local support was lacking: "Well, I haven't [received support]. My son. . . I mean, he's wonderful, but he's far away. I have really no family or friends close by" (1926–06, wife). Caregivers said they needed to learn to ask for and accept help from others. "Asking for help, that's important. And that's something it took me a little while to do. I couldn't ask friends to help. But towards the end, I did" (1927–35, wife). This can involve a change in mindset by caregivers: "One of the problems was, I got it in my head that nobody else could take care of him, I can take care of it. That's not true" (1926–01, wife).

Caregivers appreciated when others spent time with the individual with DLB and/or caregiver, which provided socialization and also gave caregivers a break. "We had gotten into a good routine with the people I had coming to help me. We really had it down to a system. . . We had our daughters and their spouses alternating time [and] had friends coming in" (1923–10, wife). Caregivers also noted the value of having company during the individual with DLB's final days: "My closest friend, my college roommate, flew up here to be with me. I don't know how I would have done it by myself. . . If there's someone who can be there at your side at the end and in the days that follow, make that happen" (1927–01, ex-wife). Caregivers often felt surprised by the willingness of others in their support system to be available, which created a more positive experience: "I did not expect to have as much help and support [as I did]. I did not expect my daughters and their spouses to put so many things on hold to be there" (1923–10, wife). Others emphasized that their children "even came in the middle of the night if we called them" (1924–04, wife).

Caregivers appreciated expressions of empathy and concern provided both verbally (e.g., checking-in with caregivers about their emotional well-being) or through actions (e.g., dropping off meals). This included community/informal support but also empathy and concern from healthcare workers:

> I applaud the hospice group, from the pastor right on down to the relief nurses over the weekends, because they were genuinely concerned. . .. Even now, I get cards from them, I get calls from them. They want to know how I'm doing, is there anything they can do for me. (1923–12, husband)

> I would ask for a replica of my experience in terms of a strong support from the hospital community, not only the physicians, but the nurse practitioners and physician's assistant, down to even the people that came in and cleaned the room; just the whole support, understanding that we needed that love and caring. Also, the nurse practitioner who followed up from neurology every three months. That continuity of support was very valuable. (1925–07, wife)

Utilizing hospice care. Some caregivers recommended trying to start hospice as early as possible, though this also had limitations, such as lacking Medicare coverage and/or losing access to non-hospice clinicians (e.g. neurologists). Multiple caregivers reported that hospice initially declined referrals for individuals with DLB who were felt to be too early in their disease to qualify. Other caregivers reported that the hospice referral came too late and everyone had to scramble. Multiple caregivers described "fabulous" and "amazing" hospice staff who showed concern for both the caregiver and the person with DLB, but experiences were not universally positive. Benefits of hospice included support for the person with DLB and their family, information about home medication administration, and explanations about what to expect.

Both positive and negative experiences with hospice revolved around the themes of communication, care coordination, quality care, and caregiver education. When health care

professionals communicated consistently and empathetically, caregivers reported a more positive experience. Lacking communication was a negative experience that made it harder for caregivers to coordinate care. Sometimes hospice helped coordinate care when the person with DLB was in a facility: "Having hospice in, I think that helped us a lot. . . It felt like we had professionals in there. . . instead of me just me constantly trying to contact her doctor and stuff, who was only available certain days and whatever" (1927–12, daughter). The degree of help coordinating care at the end of life was a driver of the end-of-life experience, both positively and negatively, with lacking coordination increasing caregiver burden: "the nurses in a way were more of a nuisance because of the constant changing times and days" (1927–59, wife).

In terms of quality care, caregivers reported dissatisfaction when health care professionals at the end of life were unfamiliar with DLB:

> I think I was surprised at how little the staff really understood the disease. And how shocked they seemed to be at her behavior. But most people there didn't have Lewy body. . . I think normally when people think of dementia, [they] assume somebody is going to be kind of withdrawn or in their own worlds or whatever, but my mom was not. She was hard to handle (1927–12, daughter).

Staff turnover and insufficient staffing also negatively affected end-of-life experiences for individuals with DLB living in facilities and their families. Caregivers described negative experiences with both under- and over-medication of the individual with DLB at the end of life. Caregivers expressed higher satisfaction when they felt hospice was providing resources such as equipment and adequate pain management. Caregivers also appreciated receiving end-of-life education from hospice staff on topics such as practical strategies (e.g. for incontinence), monitoring for medication side effects, and what to expect at the end of life.

## Discussion

The current analysis included 50 dyads of individuals with DLB and their primary informal caregivers. Individuals with DLB died 3.24 ± 1.81 years after diagnosis, typically of complications relating to the disease, though sometimes death was unexpected. Only half of caregivers reported that health care professionals discussed what to expect at the end of life in DLB and only 44% said this discussion was helpful. Most caregivers reported overall positive experiences in the last month of life of the person with DLB, as well as adequate symptom management and communication, but this was not universal. Almost 90% of individuals with DLB received hospice care, but 43% of individuals with DLB using hospice had them involved for <1 month. DLB symptoms such as impaired cognition and motor function, behavioral features, daytime sleepiness, and dependence in ADLs commonly worsened prior to death. Impaired communication, decreased appetite, and weight loss were also reported. Unexpected deaths and pre-death variability made the process more challenging for caregivers. Healthcare factors associated with positive or negative end-of-life experiences related to (1) having information about DLB and what to expect, (2) access to support, and (3) utilizing hospice care, with positive and negative hospice experiences driven by communication, care coordination, quality care, and caregiver education.

Most individuals with DLB in this analysis were men and most caregivers were women, consistent with many prior studies in DLB [21]. The mean survival from diagnosis to death was similar to prior caregiver reports in DLB [4] and only slightly lower than the disease duration reported in a meta-analysis (4.11 ± 4.10 years) [2]. Also similar to the prior survey, most

caregivers reported that the person with DLB died of complications of the disease, ascribing death to DLB itself, failure to thrive, trouble swallowing, and complications from a fall, with medical comorbidities serving as the cause of death in other circumstances [4]. This also is in line with DLB studies with more formal assessments of cause of death, where common causes of death in DLB include the dementia itself, failure to thrive, respiratory causes (e.g. pneumonia, aspiration), cardiac/circulatory causes, and multi-system organ dysfunction [22–24].

Prognostication and timely recognition of dying is one of 11 domains for optimal dementia palliative care [25], but recognizing end-of-life can be challenging for health care providers and likely requires both experience and a systematic approach [26]. Medicare guidelines for hospice use have poor sensitivity and predictive ability and thus are an inadequate guide for assessing impending end of life in dementia [27,28]. The current study identified that potential clues to approaching end of life in DLB include worsening of DLB-related symptoms (e.g. cognition, fluctuations, mobility, psychosis, behavior, daytime sleepiness), increased dependence in ADLs, decreased communication, more social withdrawal, loss of appetite, weight loss, and at the very end, breathing changes. This is consistent with caregiver interviews in a prior DLB study [5]. Additionally, increased confusion, urinary incontinence, pain, low mood, loss of appetite, and dyspnea are also described as end-of-life symptoms in dementia generally [29–31].

Hospice use in the current study was high, similar to the prior survey, with the median hospice duration of 1–2 months also aligning with prior findings [4]. The fact that 20% of participants with DLB received hospice services for <1 week is also consistent with prior results in DLB [4] and dementia more generally, where 30% of Medicare beneficiaries with dementia received hospice only in the last 7 days of life [32]. This represents an opportunity for improvement, as expert opinion and research suggest that palliative care should be provided for at least several months to achieve its full benefits [33]. Long hospice stays (6+ months) were observed for some study participants. This is consistent with prior findings that hospice stays >180 days are more common in dementia (13.7%) than when hospice is used for other chronic diseases (8.5%) or cancer (5.3%) [34].

Only half of caregivers reported that health care professionals discussed what to expect at the end of life in DLB, a marginal improvement from prior research [4] and a clear remaining quality gap. Given the unexpected nature of several deaths as reported by caregivers in both the descriptive and qualitative parts of the study, these discussions should also likely happen earlier in the disease, something recommended by caregivers in other research aiming to improve end-of-life dementia care [35]. In current interviews, caregivers described a desire for more information about DLB in general and about end-of-life in DLB in particular. This is consistent with prior research in DLB [4,5,19] and other dementias [36–39], with several prior studies suggesting that families can be unaware that dementia can be terminal [38,40,41]. Based on current and prior findings [4,5,18,19], discussions about end of life in DLB should include information regarding the fact that people with DLB often die of complications of the disease, symptoms that may herald approaching end of life in DLB, the potential unpredictability of death in DLB (including "false alarms"/fluctuations and sudden death), the role of palliative care and hospice, challenges that may be faced by caregivers (including lack of knowledge of non-specialist health care professionals), and the importance of caregiver support.

In addition to desiring more information about DLB and what to expect, caregivers in this study described wanting caregiving advice and tips. This emphasizes the importance of interdisciplinary care (e.g. occupational therapy [42]), community referrals (e.g. to caregiver training [43,44], local agencies on aging), online resources (e.g. to disease-specific advocacy organizations), and support group connections, particularly in settings where physicians/

advance practice providers have limited knowledge about caregiving practicalities. Clinicians and other team members (e.g. nurses, social workers, case management) should also encourage caregivers to seek family and community support where available and accept help from others. When families are present at medical visits, clinicians can describe the importance of family and community support for the person with DLB and their primary informal caregiver (though additional family members at clinical visits are already likely to be engaged and may need less encouragement than other potential supports).

Research shows that hospice improves the quality of life and quality of care for individuals living with dementia and their caregivers [45–47]. Most individuals in the current study used hospice and caregivers reported very good or excellent overall care, the right amount of symptom management, and respectful and goal-congruent care. Similar to prior caregiver interviews in DLB [5] and the current Last Month of Life survey results, the hospice experiences relayed in study interviews were generally but not universally positive. Both positive and negative experiences revolved around the themes of communication, care coordination, quality care, and education, with caregivers reporting more positive experiences when hospice teams were well-organized, communicative, empathetic, knowledgeable about DLB, and supportive (in terms of supplies and education). These align with many of the domains for quality palliative care in dementia, including person-centered care, communication, and shared decision-making, family care and involvement, education of the health care team, prognostication and timely recognition of dying, avoiding overly aggressive, burdensome, or futile treatment, optimal treatment of symptoms and providing comfort, and psychosocial and spiritual support [25].

Strengths of this study include the mixed-methods approach, inclusion of dyads with and without specialty DLB care, and systematic prospective enrollment of caregivers prior to the death of the person with DLB. Results will not be fully generalizable, however. The study was U.S.-based and health systems and end of life experiences may be different in other countries. Consistent with most current studies in DLB, this cohort lacked racial and ethnic diversity. This limits generalizability to other cultures and backgrounds, which has particular implications for end-of-life care [48]. DLB diagnoses were clinician-, patient-, and/or caregiver-reported and supported with a positive LBCRS, but pathologic diagnoses were not routinely available. Pathologic information is also not available during in-life clinical care, though, so current findings will be applicable to individuals with clinical diagnoses of DLB. While hospice use was sometimes brief, overall hospice utilization was high (90%). This may reflect the fact that 50% of participants reported DLB, dementia, or movement disorders specialty care and/ or that caregivers enrolled in the study were already highly engaged in their loved one's care and/or advocacy organizations. This also has implications for generalizability. Cause of death was caregiver-reported; official death certificates were not obtained. Symptoms heralding end of life in DLB were also elicited through caregiver interview. Future analyses should assess this more objectively (something planned with the ongoing PACE-DLB cohort). This study focused on the qualitative experiences of end of life in DLB, but future analyses (particularly with a higher "n") should investigate drivers of positive end-of-life experiences.

## Conclusion

While most caregivers of individuals who died with DLB reported positive end of life experiences, there are multiple opportunities for improvement. Clinicians should initiate discussions about end of life prior to advanced DLB stages. Clinicians should also provide education regarding symptoms that may herald end of life, the potential unpredictability of death, causes of death, palliative care and hospice roles, challenges faced by caregivers, and the importance

of caregiver support. Clinicians can provide interdisciplinary care and community referrals to give caregivers opportunities to learn practical strategies to aid in the end-of-life period. Clinicians need to actively monitor individuals with DLB for signs of approaching end of life and initiate timely hospice referrals. While hospice experiences were generally positive, there are also opportunities to improve hospice care (e.g. better coordination and communication). Future research should identify changes that herald end of life in DLB from a quantitative perspective (rather than post-death caregiver interviews), develop tools that can assist clinicians in evaluating disease stage to better inform patient and caregiver counseling, and quantitatively assess drivers of positive end-of-life experiences.

## Supporting information

**S1 Table. Descriptive data results.** This spreadsheet provides coded participant-level descriptive quantitative data presented in the manuscript. Participants with ages 90 or older have an entry of 90 to protect personal health information.
(XLSX)

**S1 File. End of life semi-structured interview questions.** This file provides the semi-structured interview used in the study. The current analysis presents mixed-methods data pertaining to interview question #1.
(PDF)

**S2 File. COREQ 32-item checklist.** This file provides the COREQ checklist for reporting qualitative data.
(DOCX)

**S3 File. Qualitative analysis codebook.** This file provides the qualitative analysis codebook for the current analysis including the primary contributing transcript excerpts to supplement the quotes provided in the manuscript.
(DOCX)

## Author Contributions

**Conceptualization:** Susan M. Maixner, Henry L. Paulson, Julie A. Fields, Angela Lunde, Bradley F. Boeve, James E. Galvin, Angela S. Taylor, Zhigang Li, Melissa J. Armstrong.

**Data curation:** Brian LaBarre, Zhigang Li.

**Formal analysis:** Easton Wollney, Brian LaBarre, Zhigang Li, Melissa J. Armstrong.

**Funding acquisition:** Zhigang Li, Melissa J. Armstrong.

**Investigation:** Kaitlin Sovich, Hannah J. Fechtel, Melissa J. Armstrong.

**Methodology:** Melissa J. Armstrong.

**Project administration:** Kaitlin Sovich, Susan M. Maixner, Henry L. Paulson, Carol Manning, Julie A. Fields, Angela Lunde, Bradley F. Boeve, James E. Galvin, Angela S. Taylor, Zhigang Li, Hannah J. Fechtel, Melissa J. Armstrong.

**Supervision:** Susan M. Maixner, Henry L. Paulson, Carol Manning, Julie A. Fields, Angela Lunde, Bradley F. Boeve, James E. Galvin, Zhigang Li, Melissa J. Armstrong.

**Writing – original draft:** Easton Wollney, Melissa J. Armstrong.

**Writing – review & editing:** Easton Wollney, Kaitlin Sovich, Brian LaBarre, Susan M. Maix-ner, Henry L. Paulson, Carol Manning, Julie A. Fields, Angela Lunde, Bradley F. Boeve, James E. Galvin, Angela S. Taylor, Zhigang Li, Hannah J. Fechtel, Melissa J. Armstrong.

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
