## [Decision Letter · Decision Letter 0]

16 Jun 2024

PONE-D-24-08238End-of-Life Experiences in Individuals with Dementia with Lewy Bodies: a Mixed-Methods AnalysisPLOS ONE

Dear Dr. Armstrong,

Thank you for submitting your manuscript to PLOS ONE. After careful consideration, we feel that it has merit but does not fully meet PLOS ONE’s publication criteria as it currently stands. Therefore, we invite you to submit a revised version of the manuscript that addresses the points raised during the review process. 

**Reviewer 1**

Major Revision

Thank you for submitting an interesting manuscript that explores caregiver-reported end-of-life experiences in DLB. Please consider the following suggestions for improvement by section.

Title:

- Although the title references “End-of-Life Experiences in Individuals with Dementia with Lewy Bodies,” the majority of data are caregiver-reported and may not accurately reflect the patient experience. To avoid confusion, consider incorporating caregivers into the title (e.g., Caregiver-Reported End-of-Life Experiences in Individuals with DLB, End-of-Life Experiences among DLB Patients and Caregivers, Patient and Caregiver Experiences at End-of-Life in DLB, etc.)

Abstract:

- In the methods section, please clarify what data was collected from patients, if any

Introduction:

- You state: “Reported survival is longer in pathologic cohorts.” Please define the term “pathologic cohorts” for those that are unfamiliar. I assume these were patients who underwent post-mortem examination, but this was not immediately clear.

- You state, “Allowing for multiple categories, the most common cause of death in DLB is failure to thrive (65%), followed by pneumonia/swallowing difficulties (23%), medical conditions other than pneumonia (19%), and falls or complications from a fall (10%). How big was this cohort? The denominator will help put these figures into context, as death certificates are often unavailable when using national data.

- Similarly, when you state “In a 2017 survey conducted through the Lewy Body Dementia Association, only 40% of family members of individuals with DLB said their physician discussed what to expect at the end of life in DLB,” what was the sample size?

- You report that “The current mixed-methods analysis investigates end-of-life experiences for the first 50 study participants who died during the study.” Why are you reporting these data separately from the parent study?

Methods:

- The inclusion criteria includes persons with DLB expected to live at least 6 months. How was this determined?

Results:

- You present mean years between diagnosis and death. It would be helpful to present data on disease duration (symptom onset to death) if available.

- There were 37 male patients and 36 patients who identified as male. Was one patient transgender? Consider changing sex and gender to “biological sex” and “gender identity” to avoid confusion.

- Caregivers most commonly reported that individuals with DLB died from the DLB itself. How? Please elaborate.

- In the section “Healthcare factors associated with Positive or Negative Experiences,” please describe the negative factors in more detail. For example, what were the negative experiences surrounding hospice care?

Discussion:

- A 90% hospice utilization rate is quite high, even if initiated later than appropriate. Could the high utilization rate be based on referral patterns from outpatient doctors (more than 50% saw a DLB specialist) or due to participant bias, given that data were drawn from patients who were willing to participate in end-of-life care research? I would add this to study limitations since this affects generalizability.

General:

- Writing could be more concise, particularly in the introduction

**Reviewer 2**

Minor Revision

The paper is good and interesting but needs minor revision. See my comments below

Abstract

It does not have conclusion which should be strictly based on the results

Discussion should be deleted from the abstract

Introduction

DLB expand to full at the beginning of the statement

Study visits and measures

Correct priori

Please be specific of what you did in the research-visits were completed over multiple

days if needed

Data Availability

Indicate the author to contact for data availability

Sample size of 50

It is small, you would have followed more subjects

Conclusion

Remove the table you mentioned on the conclusion. The conclusion is too long. Kindly summarise the key findings and implications

We look forward to receiving your revised manuscript.

Kind regards,

Hope Onohuean, PhD

Academic Editor

PLOS ONE

Journal Requirements:

[I have read the journal's policy and the authors of this manuscript have the following competing interests: EW, BL, SMM, ZL, and HJF report no competing interests. HLP: HLP receives funding from the NIA (1P30AG053760) and is the local PI of a Lewy Body Dementia Association Research Center of Excellence. CM: CM receives research support from ACL/DHHS (90ALGG0014-01-00), NIA/NIH (2SB1AG037357-04A1, R01-AG-054435), HRSA (U1QHP287440400) and DoD (AZ190036). She is the local PI of a Lewy Body Dementia Association Research Center of Excellence. JAF: JAF receives research support from the NIA (U01NS100620, R01AG068128, R43AG65088). AL: AL receives research support from the NIA (P30AG62677, R43AG65088). She is a Program Coordinator for the local a Lewy Body Dementia Association Coordinating Center Research Center of Excellence. BFB: BFB has served as an investigator for clinical trials sponsored by Biogen, Alector, and EIP Pharma. He serves on the Scientific Advisory Board of the Lewy Body Dementia Association, Association for Frontotemporal Degeneration, and Tau Consortium. He is the site PI of a Lewy Body Dementia Association Research Center of Excellence program, as well as coordinating center PI of the program. He receives research support from the NIH, the Mayo Clinic Dorothy and Harry T. Mangurian Jr. Lewy Body Dementia Program, and the Little Family Foundation. JEG: JEG is the creator of the QDRS and the LBCRS. He is supported by grants from the National Institutes of Health (R01 AG069765, R01 AG057681, R01 NS101483, P30 AG059295, U54 AG06354, R01 AG056531, U01 NS100610, R01 AG056610, R01 AG054425, R01 AG068128) and the Leo and Anne Albert Charitable Trust. He is the local PI of the Lewy Body Dementia Association Research Center of Excellence at the University of Miami and serves on the Scientific Advisory Board of the Lewy Body Dementia Association. AST: AST is an employee of the Lewy Body Dementia Association. MJA: MJA receives research support from the NIH (R01AG068128, P30AG066506, R01NS121099, R44AG062072), the Florida Department of Health (grants 20A08, 24A14, 24A15), and as the local PI of a Lewy Body Dementia Association Research Center of Excellence. She serves on the DSMBs for the Alzheimer’s Therapeutic Research Institute/Alzheimer’s Clinical Trial Consortium and the Alzheimer’s Disease Cooperative Study. She has provided educational content for Medscape, Vindico CME, and Prime Inc.]. 

Reviewers' comments:

Reviewer's Responses to Questions

**Comments to the Author**

1. Is the manuscript technically sound, and do the data support the conclusions?

Reviewer #1: Yes

Reviewer #2: Yes

2. Has the statistical analysis been performed appropriately and rigorously? 

Reviewer #1: Yes

Reviewer #2: Yes

3. Have the authors made all data underlying the findings in their manuscript fully available?

Reviewer #1: No

Reviewer #2: Yes

4. Is the manuscript presented in an intelligible fashion and written in standard English?

Reviewer #1: Yes

Reviewer #2: Yes

5. Review Comments to the Author

Reviewer #1: Thank you for submitting an interesting manuscript that explores caregiver-reported end-of-life experiences in DLB. Please consider the following suggestions for improvement by section.

Title:

- Although the title references “End-of-Life Experiences in Individuals with Dementia with Lewy Bodies,” the majority of data are caregiver-reported and may not accurately reflect the patient experience. To avoid confusion, consider incorporating caregivers into the title (e.g., Caregiver-Reported End-of-Life Experiences in Individuals with DLB, End-of-Life Experiences among DLB Patients and Caregivers, Patient and Caregiver Experiences at End-of-Life in DLB, etc.)

Abstract:

- In the methods section, please clarify what data was collected from patients, if any

Introduction:

- You state: “Reported survival is longer in pathologic cohorts.” Please define the term “pathologic cohorts” for those that are unfamiliar. I assume these were patients who underwent post-mortem examination, but this was not immediately clear.

- You state, “Allowing for multiple categories, the most common cause of death in DLB is failure to thrive (65%), followed by pneumonia/swallowing difficulties (23%), medical conditions other than pneumonia (19%), and falls or complications from a fall (10%). How big was this cohort? The denominator will help put these figures into context, as death certificates are often unavailable when using national data.

- Similarly, when you state “In a 2017 survey conducted through the Lewy Body Dementia Association, only 40% of family members of individuals with DLB said their physician discussed what to expect at the end of life in DLB,” what was the sample size?

- You report that “The current mixed-methods analysis investigates end-of-life experiences for the first 50 study participants who died during the study.” Why are you reporting these data separately from the parent study?

Methods:

- The inclusion criteria includes persons with DLB expected to live at least 6 months. How was this determined?

Results:

- You present mean years between diagnosis and death. It would be helpful to present data on disease duration (symptom onset to death) if available.

- There were 37 male patients and 36 patients who identified as male. Was one patient transgender? Consider changing sex and gender to “biological sex” and “gender identity” to avoid confusion.

- Caregivers most commonly reported that individuals with DLB died from the DLB itself. How? Please elaborate.

- In the section “Healthcare factors associated with Positive or Negative Experiences,” please describe the negative factors in more detail. For example, what were the negative experiences surrounding hospice care?

Discussion:

- A 90% hospice utilization rate is quite high, even if initiated later than appropriate. Could the high utilization rate be based on referral patterns from outpatient doctors (more than 50% saw a DLB specialist) or due to participant bias, given that data were drawn from patients who were willing to participate in end-of-life care research? I would add this to study limitations since this affects generalizability.

General:

- Writing could be more concise, particularly in the introduction

Reviewer #2: The paper is good and interesting but needs minor revision. See my comments below

Abstract

It does not have conclusion which should be strictly based on the results

Discussion should be deleted from the abstract

Introduction

DLB expand to full at the beginning of the statement

Study visits and measures

Correct priori

Please be specific of what you did in the research-visits were completed over multiple

days if needed

Data Availability

Indicate the author to contact for data availability

Sample size of 50

It is small, you would have followed more subjects

Conclusion

Remove the table you mentioned on the conclusion. The conclusion is too long. Kindly summarise the key findings and implications

6. PLOS authors have the option to publish the peer review history of their article (what does this mean?). If published, this will include your full peer review and any attached files.

Reviewer #1: No

Reviewer #2: No

---

## [Author Response · Author response to Decision Letter 0]

30 Jul 2024

Response to Reviewers for PONE-D-24-08238, End-of-Life Experiences in Individuals with Dementia with Lewy Bodies: a Mixed-Methods Analysis

Reviewer 1

Thank you for submitting an interesting manuscript that explores caregiver-reported end-of-life experiences in DLB. Please consider the following suggestions for improvement by section.

Title:

- Although the title references “End-of-Life Experiences in Individuals with Dementia with Lewy Bodies,” the majority of data are caregiver-reported and may not accurately reflect the patient experience. To avoid confusion, consider incorporating caregivers into the title (e.g., Caregiver-Reported End-of-Life Experiences in Individuals with DLB, End-of-Life Experiences among DLB Patients and Caregivers, Patient and Caregiver Experiences at End-of-Life in DLB, etc.)

Author Response: We have changed the title to “End-of-Life Experiences in Individuals with Dementia with Lewy Bodies and their Caregivers: a Mixed-Methods Analysis.”

Abstract:

- In the methods section, please clarify what data was collected from patients, if any

Author Response: We have revised the methods section in the abstract to read, “Dyads of individuals with moderate-advanced DLB and their primary informal caregivers were recruited from specialty clinics, advocacy organizations, and research registries and followed prospectively every 6 months. The current study examines results of caregiver study visits 3 months after the death of the person with DLB. These visits included the Last Month of Life survey, study-specific questions, and a semi-structured interview querying end-of-life experiences.” (We also made edits to comply with the 300 word count limit.) Data was from the post-death visit completed by the caregiver 3 months after the death of the person with DLB. We have also added this clarification to the last paragraph of the introduction (page 5): “The current mixed-methods analysis investigates end-of-life experiences for the first 50 study participants who died during the study as reported by caregivers at the 3-month post-death visit.”

Introduction:

- You state: “Reported survival is longer in pathologic cohorts.” Please define the term “pathologic cohorts” for those that are unfamiliar. I assume these were patients who underwent post-mortem examination, but this was not immediately clear.

Author Response: We have changed the wording to, “Reported survival is longer in pathologic cohorts (i.e., with post-mortem examination) [3]…” (page 4)

- You state, “Allowing for multiple categories, the most common cause of death in DLB is failure to thrive (65%), followed by pneumonia/swallowing difficulties (23%), medical conditions other than pneumonia (19%), and falls or complications from a fall (10%). How big was this cohort? The denominator will help put these figures into context, as death certificates are often unavailable when using national data.

Author Response: We have revised this sentence to read, “Based on a prior survey (n=658), the most common cause of death in DLB is failure to thrive (65%), followed by pneumonia/swallowing difficulties (23%), medical conditions other than pneumonia (19%), and falls or complications from a fall (10%) (multiple categories allowed) [4].” (page 4)

- Similarly, when you state “In a 2017 survey conducted through the Lewy Body Dementia Association, only 40% of family members of individuals with DLB said their physician discussed what to expect at the end of life in DLB,” what was the sample size?

Author Response: We have added the “n” from the prior study so that the sentence now reads: “In a 2017 survey conducted through the Lewy Body Dementia Association (n=658), only 40% of family members of individuals with DLB said their physician discussed what to expect at the end of life in DLB. Only 22% reported this was discussed to a helpful degree [4].” (page 4)

- You report that “The current mixed-methods analysis investigates end-of-life experiences for the first 50 study participants who died during the study.” Why are you reporting these data separately from the parent study?

Author Response: The parent longitudinal study is looking at patient and caregiver experiences in moderate-advanced DLB, which involves a broad assessment every 6 months. The current analysis focuses only on results of the 3-month post-death caregiver experience, which is its own study aim. Additionally, the current mixed-methods analysis reports specifically on caregivers who completed the original semi-structured interview, which subsequently changed since saturation of themes was reached. We have revised the study participant section in the methods to clarify this: “The current analysis includes the first 50 dyads with a post-death study visit. Because saturation of themes was reached with this cohort, the interview questions subsequently changed. Thus, the current analysis includes visits relating to the original semi-structured interview. Included post-death visits reflect participants who died after the baseline visit and before the 6-month follow up and participants who had multiple study visits prior to death.” (page 6) We also added this information: “The semi-structured interview changed after this time due to saturation of themes for all interview questions.” (page 7) To the study visits and measures section, we added the clarification “Visits occurred every 6 months until the death of the person with DLB. A post-death visit was completed with the caregiver approximately 3 months after the death of the person with DLB and this visit formed the basis for the current analysis” (page 7, italics = new).

Methods:

- The inclusion criteria includes persons with DLB expected to live at least 6 months. How was this determined?

Author Response: We have revised this to clarify: “…5) person with DLB expected to live at least 6 months (by clinician or participant estimation)…” (page 6)

Results:

- You present mean years between diagnosis and death. It would be helpful to present data on disease duration (symptom onset to death) if available.

Author Response: We only collected year of diagnosis, not year of symptom onset.

- There were 37 male patients and 36 patients who identified as male. Was one patient transgender? Consider changing sex and gender to “biological sex” and “gender identity” to avoid confusion.

Author Response: We asked sex and gender separately, but did not ask participants on additional details how they view their identity. It is correct that one person identified a different gender than sex. We have changed the terms to “Biologic Sex” and “Gender Identity” as requested (Table 1).

- Caregivers most commonly reported that individuals with DLB died from the DLB itself. How? Please elaborate.

Author Response: This was caregiver-reported and details were not provided. We edited the sentence to read, “Caregivers most commonly reported that individuals with DLB died from the DLB itself (21, 42%; exact details unknown) or failure to thrive (9, 18%).” (page 12)

- In the section “Healthcare factors associated with Positive or Negative Experiences,” please describe the negative factors in more detail. For example, what were the negative experiences surrounding hospice care?

Author Response: We have made revisions throughout this section (page 25-26) to specifically call out the negative experiences. Examples of calling out the negative experiences, “Lacking communication was a negative experience that made it harder for caregivers to coordinate care.” (page 25). “The degree of help coordinating care at the end of life was a driver of the end-of-life experience, both positively and negatively, with lacking coordination increasing caregiver burden: “the nurses in a way were more of a nuisance because of the constant changing times and days” (1927-59, wife).” (page 25) “Staff turnover and insufficient staffing also negatively affected end-of-life experiences for individuals with DLB living in facilities and their families. Caregivers described negative experiences with both under- and over-medication of the individual with DLB at the end of life.” (page 26)

Discussion:

- A 90% hospice utilization rate is quite high, even if initiated later than appropriate. Could the high utilization rate be based on referral patterns from outpatient doctors (more than 50% saw a DLB specialist) or due to participant bias, given that data were drawn from patients who were willing to participate in end-of-life care research? I would add this to study limitations since this affects generalizability.

Author Response: We have added this to the limitation section (page 30-31): “While hospice use was sometimes brief, overall hospice utilization was high (90%). This may reflect the fact that 50% of participants reported DLB, dementia, or movement disorders specialty care and/or that caregivers enrolled in the study were already highly engaged in their loved one’s care and/or advocacy organizations. This also has implications for generalizability.”

General:

- Writing could be more concise, particularly in the introduction

Author Response: We have edited the introduction to take out some sentences that were less important (e.g., we removed “DLB is one of two dementias under the Lewy body dementia umbrella”) and to make other sentences less wordy. 

Reviewer 2

The paper is good and interesting but needs minor revision. See my comments below

Abstract: It does not have conclusion which should be strictly based on the results. Discussion should be deleted from the abstract.

Author Response: We have changed the header of the final section from discussion to conclusions. We have also edited the wording to show the conclusions tie to the results. At the same time we have edited other parts of the abstract to make sure it is not over the 300 word limit.

Introduction: DLB expand to full at the beginning of the statement

Author Response: Dementia with Lewy bodies is written out in full in the first sentence of the introduction.

Study visits and measures: Correct priori. Please be specific of what you did in the research-visits were completed over multiple. days if needed.

Author Response: We are uncertain what “priori” needs to be corrected. We reviewed all occurrences of “priori” and “prior” in the manuscript. The sentence “Study visits were designed for virtual and in-person completion given a priori plans for inclusion of a virtual cohort” (page 7) appears correct and this is the only occurrence of “priori.” Where “prior” is used this is what is intended (i.e., we did not use “prior” when intending “priori”). We have removed the sentence about completing research visits over multiple days because this was not relevant for the post-death caregiver study visits. All post-death caregiver visits except one were completed in one sitting.

Data Availability: Indicate the author to contact for data availability

Author Response: We have added “by the corresponding author” (page 9).

Sample size of 50. It is small, you would have followed more subjects.

Author Response: The parent PACE-DLB is studying more participants. The current analysis looks at the first 50 participants who died because saturation of themes for the interview questions was achieved with this sample size (which is actually larger than most qualitative studies). We have now clarified this in a couple places: “The current analysis includes the first 50 dyads with a post-death study visit. Because saturation of themes was reached with this cohort, the interview questions subsequently changed. Thus, the current analysis includes visits relating to the original semi-structured interview. Included post-death visits reflect participants who died after the baseline visit and before the 6-month follow up and participants who had multiple study visits prior to death.” (page 6) We also added this information to the methods: “The semi-structured interview changed after this time due to saturation of themes for all interview questions.” (page 7) 

Conclusion: Remove the table you mentioned on the conclusion. The conclusion is too long. Kindly summarise the key findings and implications.

Author Response: We have deleted Table 4 as requested. We have also edited the paragraph to make it more concise.

RESPONSES TO EDITOR COMMENTS (see also cover letter):

We reviewed the PLoS One formatting and adjusted the manuscript to match. 

As requested we updated the competing interests statement with the additional sentence at the end (no other changes):

I have read the journal's policy and the authors of this manuscript have the following competing interests: EW, BL, SMM, ZL, and HJF report no competing interests. HLP: HLP receives funding from the NIA (1P30AG053760) and is the local PI of a Lewy Body Dementia Association Research Center of Excellence. CM: CM receives research support from ACL/DHHS (90ALGG0014-01-00), NIA/NIH (2SB1AG037357-04A1, R01-AG-054435), HRSA (U1QHP287440400) and DoD (AZ190036). She is the local PI of a Lewy Body Dementia Association Research Center of Excellence. JAF: JAF receives research support from the NIA (U01NS100620, R01AG068128, R43AG65088). AL: AL receives research support from the NIA (P30AG62677, R43AG65088). She is a Program Coordinator for the local a Lewy Body Dementia Association Coordinating Center Research Center of Excellence. BFB: BFB has served as an investigator for clinical trials sponsored by Biogen, Alector, and EIP Pharma. He serves on the Scientific Advisory Board of the Lewy Body Dementia Association, Association for Frontotemporal Degeneration, and Tau Consortium. He is the site PI of a Lewy Body Dementia Association Research Center of Excellence program, as well as coordinating center PI of the program. He receives research support from the NIH, the Mayo Clinic Dorothy and Harry T. Mangurian Jr. Lewy Body Dementia Program, and the Little Family Foundation. JEG: JEG is the creator of the QDRS and the LBCRS. He is supported by grants from the National Institutes of Health (R01 AG069765, R01 AG057681, R01 NS101483, P30 AG059295, U54 AG06354, R01 AG056531, U01 NS100610, R01 AG056610, R01 AG054425, R01 AG068128) and the Leo and Anne Albert Charitable Trust. He is the local PI of the Lewy Body Dementia Association Research Center of Excellence at the University of Miami and serves on the Scientific Advisory Board of the Lewy Body Dementia Association. AST: AST is an employee of the Lewy Body Dementia Association. MJA: MJA receives research support from the NIH (R01AG068128, P30AG066506, R01NS121099, R44AG062072), the Florida Department of Health (grants 20A08, 24A14, 24A15), and as the local PI of a Lewy Body Dementia Association Research Center of Excellence. She serves on the DSMBs for the Alzheimer’s Therapeutic Research Institute/Alzheimer’s Clinical Trial Consortium and the Alzheimer’s Disease Cooperative Study. She has provided educational content for Medscape, Vindico CME, and Prime Inc. This does not alter our adherence to PLOS ONE policies on sharing data and materials.

With regard to data availability, we viewed this as primarily a qualitative study and limited data sharing because the interviews frequently contain identifying and sensitive data. However, in rereading the PLoS One data sharing instructions, it is permissible to share “excerpts of the transcripts relevant to the study” and not full interviews. We have now thus shared data in two files for the supporting information. The study doesn’t include much demographic/descriptive data, but what is presented is now included in S3 Table. The codebook with relevant excerpts is now S4 File. We’ve changed our data availability paragraph in the manuscript to read, “Individual participant demographic data and descriptive quantitative data presented in this manuscript are available in the S3 Table. The codebook with relevant excerpts is available in S4 File to supplement the quotes provided in the manuscript.”

As requested, we have provided captions for the supporting information files. We have also reviewed the reference list and provide appropriate formatting in the “clean” version of the revised manuscript.

---

## [Decision Letter · Decision Letter 1]

13 Aug 2024

End-of-Life Experiences in Individuals with Dementia with Lewy Bodies and their Caregivers: a Mixed-Methods Analysis

PONE-D-24-08238R1

Dear Dr Melissa J. Armstrong.

Thank you for addressing the comments and resubmitting your interesting manuscript that explores caregiver-reported end-of-life experiences in DLB.

We’re pleased to inform you that your manuscript has been judged scientifically suitable for publication and will be formally accepted for publication once it meets all outstanding technical requirements.

Kind regards,

Hope Onohuean, PhD

Academic Editor

PLOS ONE

Additional Editor Comments (optional):

Reviewers' comments:

Reviewer's Responses to Questions

**Comments to the Author**

1. If the authors have adequately addressed your comments raised in a previous round of review and you feel that this manuscript is now acceptable for publication, you may indicate that here to bypass the “Comments to the Author” section, enter your conflict of interest statement in the “Confidential to Editor” section, and submit your "Accept" recommendation.

Reviewer #1: All comments have been addressed

Reviewer #2: All comments have been addressed

2. Is the manuscript technically sound, and do the data support the conclusions?

Reviewer #1: Yes

Reviewer #2: Yes

3. Has the statistical analysis been performed appropriately and rigorously? 

Reviewer #1: Yes

Reviewer #2: Yes

4. Have the authors made all data underlying the findings in their manuscript fully available?

Reviewer #1: Yes

Reviewer #2: Yes

5. Is the manuscript presented in an intelligible fashion and written in standard English?

Reviewer #1: Yes

Reviewer #2: Yes

6. Review Comments to the Author

Reviewer #1: (No Response)

Reviewer #2: The comments have been addressed from the concerns raised in the reviews. The paper is interesting and highly recommnded

7. PLOS authors have the option to publish the peer review history of their article (what does this mean?). If published, this will include your full peer review and any attached files.

Reviewer #1: No

Reviewer #2: **Yes: **Emmanuel Ifeanyi Obeagu

---

## [Editor Report · Acceptance letter]

20 Aug 2024

PONE-D-24-08238R1 

PLOS ONE

Dear Dr. Armstrong, 

I'm pleased to inform you that your manuscript has been deemed suitable for publication in PLOS ONE. Congratulations! Your manuscript is now being handed over to our production team.

Kind regards, 

on behalf of

Dr. Hope Onohuean 

Academic Editor

PLOS ONE